# Post-acute COVID-19 associated with evidence of bystander T-cell activation and a recurring antibiotic-resistant bacterial pneumonia

Michaela Gregorova[1†], Daniel Morse[1†], Tarcisio Brignoli[1†], Joseph Steventon[1], Fergus Hamilton[2], Mahableshwar Albur[2], David Arnold[2], Matthew Thomas[2], Alice Halliday[1], Holly Baum[1], Christopher Rice[1], Matthew B Avison[1], Andrew D Davidson[1], Marianna Santopaolo[1], Elizabeth Oliver[1], Anu Goenka[1], Adam Finn[1], Linda Wooldridge[3], Borko Amulic[1], Rosemary J Boyton[4,5], Daniel M Altmann[4], David K Butler[4], Claire McMurray[6], Joanna Stockton[6], Sam Nicholls[6], Charles Cooper[6], Nicholas Loman[6], Michael J Cox[5], Laura Rivino[1‡*], Ruth C Massey[1‡*]

[1]School of Cellular and Molecular Medicine, University of Bristol, Bristol, United Kingdom; [2]North Bristol NHS Trust, Bristol, United Kingdom; [3]Bristol Veterinary School in the Faculty of Health Sciences, Bristol, United Kingdom; [4]Department of Infectious Disease, Imperial College London, London, United Kingdom; [5]Lung Division, Royal Brompton & Harefield NHS Foundation Trust, London, United Kingdom; [6]Institute of Microbiology and Infection, University of Birmingham, Birmingham, United Kingdom

*For correspondence:
laura.rivino@bristol.ac.uk (LR);
ruth.massey@bristol.ac.uk (RCM)

[†]These authors contributed equally to this work
[‡]These authors also contributed equally to this work

Competing interests: The authors declare that no competing interests exist.

**Abstract** Here, we describe the case of a COVID-19 patient who developed recurring ventilator-associated pneumonia caused by *Pseudomonas aeruginosa* that acquired increasing levels of antimicrobial resistance (AMR) in response to treatment. Metagenomic analysis revealed the AMR genotype, while immunological analysis revealed massive and escalating levels of T-cell activation. These were both SARS-CoV-2 and *P. aeruginosa* specific, and bystander activated, which may have contributed to this patient's persistent symptoms and radiological changes.

## Introduction

The COVID-19 pandemic has brought with it the largest ever cohort of patients requiring mechanical ventilation. Complications associated with such severe viral infections are many-fold, and include increased susceptibility to secondary bacterial infections (*Zhou et al., 2020*; *Langford et al., 2020*), as well as post-acute COVID-19, where patients experience symptoms extending beyond 3 weeks from the onset of their first COVID-19 symptoms (*Greenhalgh et al., 2020*). The first report of secondary infections in COVID-19 patients was from Wuhan in March 2020, where 15% of hospitalized patients developed secondary infections, and of those who did not survive their SARS-CoV-2 infection, 50% had a secondary bacterial infection (*Zhou et al., 2020*). Since then many COVID-19 studies reporting secondary infections have been published, with a recent meta-analysis of 24 independent studies that included 3338 patients from five countries reporting that 14.3% of hospitalized COVID-19 patients developed a secondary bacterial infection, which is associated with significant morbidity, mortality and the financial costs associated with prolonged hospitilisation (*Langford et al., 2020*). The incidence of post-acute COVID-19 varies depending on the group of patients considered, with approximately 10% of patients who have tested positive for SARS-CoV-2 virus remaining unwell

beyond 3 weeks (*Greenhalgh et al., 2020*). However, this can be as high as 74% when hospitalised patients are considered, where symptoms include breathlessness and excessive fatigue, with abnormal radiological features reported in 12% of this cohort (*Arnold et al., 2020*).

The DISCOVER study (DIagnostic and Severity markers of COVID-19 to Enable Rapid triage, REC: 20/YH/0121) was established in March 2020 to collect and analyse longitudinal samples from COVID-19 patients. One study participant, an otherwise healthy male between 45 and 55 years of age presented to hospital with Type-1 respiratory failure (hypoxaemia), 20 days after he tested positive for SARS-CoV-2 by RT-PCR. At the time of testing he was asymptomatic (tested as a house-hold contact of a health-care worker), and this represents Day one on the time-line presented in *Figure 1*. He became symptomatic for COVID-19 13 days after this, which is within the incubation period for SARS-CoV-2 as defined by the CDC of 14 days (*COVID-19 (Coronavirus Disease), 2020*), and his health declined over the following week. Upon admission to hospital a chest X-ray was taken, and he was admitted to the ICU where he was mechanically ventilated (*Figure 1*). An RT-PCR test on an endotracheal sample collected at this time did not detect SARS-CoV-2 suggesting that he had cleared the viral infection.

Once admitted to our intensive care unit he was assessed, as all our patients are daily, for signs of additional infection through a process led by a consultant medical microbiologist along with senior pharmacist. At these meetings, the clinical status of the patient, radiological features, oxygen requirements, inflammatory indices (WCC, CRP, and procalcitonin) and any culture results are discussed. We had no concerns at the time of admission of this patient to the ICU that he had any infection other than that of SARS-CoV-2. Computerised tomography (CT) scans of the patient were collected through his time in the ICU and the classic features of COVID-19, including ground-glass opacities, were evident throughout (*Figure 3—figure supplement 1*).

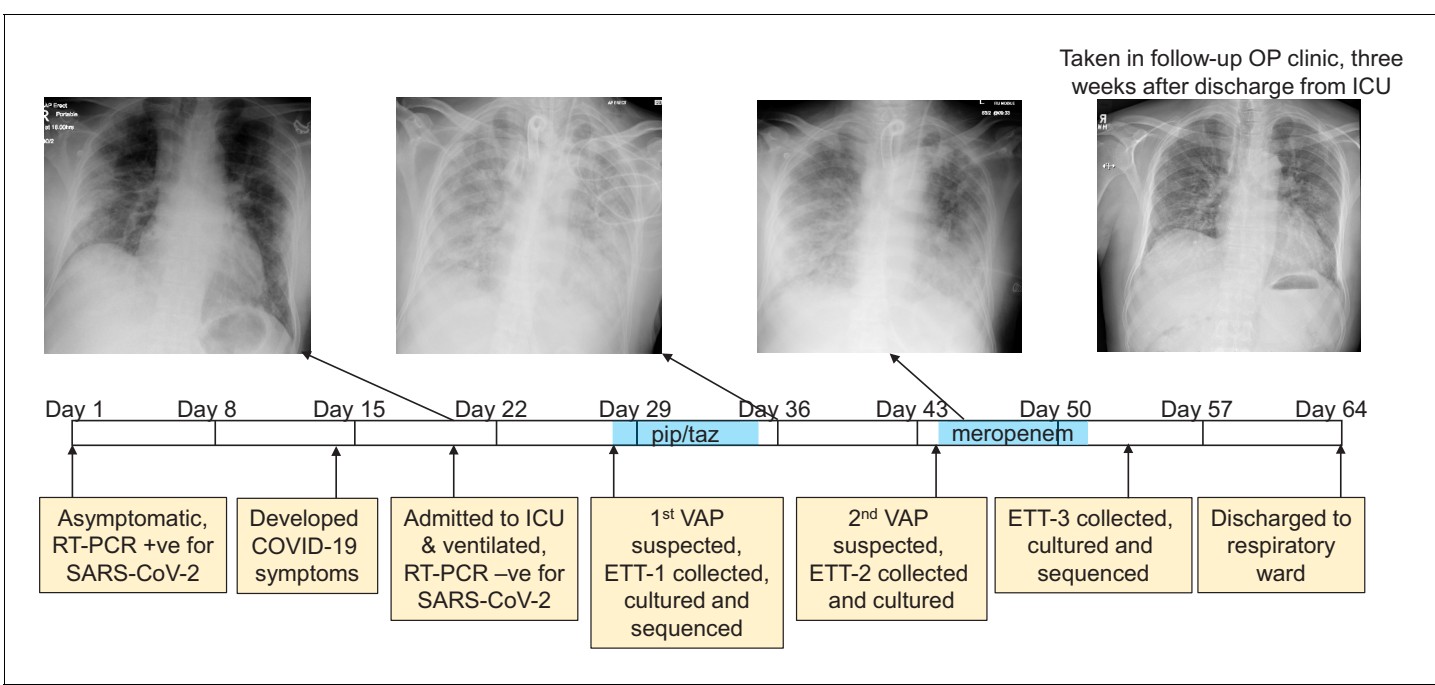

**Figure 1.** The development of a recurring ventilator-associated pneumonia (VAP) by a COVID-19 patient. A clinical time-line is presented from the point at which the patient tested positive for SARS-CoV-2 (day 1), through to his discharge from the ICU (day 64). Noteworthy clinical features are indicated in the yellow boxes below the time-line. The three x-rays taken during the patient's time in the ICU are presented, as is a later follow-up x-ray taken in an out-patient (OP) clinic 3 weeks after discharge. The time during which the antibiotics piperacillin and tazobactam (pip/taz) and meropenem were administered to the patient to treat the VAP are indicated in blue. The points at which endotracheal tube aspirates (ETT) were collected and the subsequent analysis of these also are indicated and described.

The online version of this article includes the following figure supplement(s) for figure 1:

**Figure supplement 1.** CT scans of the patient at three time points during his stay in the ICU, Days 20, 28 and 40.

After a week in the ICU he was diagnosed with ventilator-associated pneumonia (VAP) based on the clinical, radiological, biochemical, and microbiological parameters described above. An antibiotic susceptible *Pseudomonas aeruginosa* strain was cultured from an aspirate collected from his endotracheal tube (ETT-1, *Figure 1*), (minimum inhibitory concentration (MIC) meropenem 0.5 mg/l; piperacillin-tazobactam (pip/taz) 8 mg/l). He was prescribed a seven-day course of pip/taz (4.5 g every 6 hr), and clinically recovered from this bacterial infection. Eight days after finishing this first course of antibiotics his VAP recurred (diagnosed as described above), and a pip/taz resistant *P. aeruginosa* was cultured from ETT-2 (minimum inhibitory concentration (MIC) >16 mg/l). He was prescribed a seven-day course of meropenem (1 g every 8 hr) and showed signs of clinical improvement. There were no further concerns about a bacterial infection during his ICU stay (i.e. he did not meet any of the clinical diagnostic criteria described above), although a third *P. aeruginosa* was cultured from a sample (ETT-3) collected two days after he completed his course of meropenem that was resistant to both pip/taz and meropenem (MICs > 16 mg/l and >8 mg/l respectively). His health continued to improve, and he was discharged to the respiratory ward a week later. The patient attended a follow-up clinic 3 weeks after hospital discharge where he reported on-going symptoms such as breathlessness and myalgia, and there was a significant drop in his oxygen saturations after mild exertion to 84%. Eight months following his positive test he is still reporting breathlessness and fatigue, common symptoms of the newly defined 'Long COVID' condition.

To understand the cellular and molecular dynamics of this post-acute COVID-19 case from the perspective of both the pathogen and patient's immune response, longitudinal respiratory and blood samples were collected and analysed with a view to identifying early diagnostic biomarkers of infection onset and potential opportunities for immunotherapeutic intervention.

## Results

Metagenomics was used to characterise in depth the composition and genomic features of the bacteria present in this patient's lower respiratory tract from when he first developed VAP (ETT-1, *Figure 1*) to when he had recovered from his second VAP (ETT-3, *Figure 1*). ETT-2 collected during his second bout of VAP was unfortunately not available for sequencing. We extracted the entire genetic material from 500 µl of ETTs 1 and 3 with no bacterial enrichment or human DNA depletion steps, and to ensure data for the bacterial component of the samples was generated, these were sequenced on a PromethION (Oxford Nanopore technology). The bacterial DNA from ETT-1 was entirely that of *P. aeruginosa* corresponding to multi-locus sequence type ST253 (*Jolley et al., 2018*), a world-wide clone frequently associated with AMR epidemics (*Treepong et al., 2018*; *Figure 2*). There was a greater diversity of bacterial DNA in ETT-3, as the patient recovered from his SARS-CoV-2 and *P. aeruginosa* infections (*Figure 2b*), and we were able to construct a whole genome of the *P. aeruginosa* strain from within this, which again corresponded to multi-locus sequence type ST253. *P. aeruginosa* is a ubiquitous environmental bacterium, and while it can be found as a commensal in the respiratory tract of some individuals, this is typically only when they have some underlying chronic suppurative lung disease such as bronchiectasis. Prior to his COVID-19 diagnosis this relatively young patient was healthy with no prior medical issues, which suggests the *P. aeruginosa* infections were hospital or ICU acquired.

Resistance to beta-lactam antibiotics by *P. aeruginosa* can be multifactorial and includes the acquisition of single nucleotide polymorphisms (SNPs) in efflux and porin genes that affect the passage of the antibiotic into and out of the bacterial cell (*Blair et al., 2015*). Analysis of the sequence data from ETT-3 generated a whole genome for *P. aeruginosa*, again corresponding to ST253, that aligned with >99% identify to the genome from ETT-1. There were however two noteworthy SNPs in this later sample that explain the increased AMR of this isolate. The multi-drug efflux pump, MexAB, is transcriptionally regulated by the translated product of the *mexR* gene, MexR. We found a SNP in *mexR* that converts a key amino-acid (Arg91-Gln) in the DNA-binding domain of the MexR protein, that would likely result in the de-repression of this efflux system (*Saito et al., 2003*). A second SNP was found that introduced a premature stop codon (Tyr120-stop) in the gene encoding the outer membrane protein OprD, a protein with a well-established role in the entry of meropenem into the bacterial cell (*Quale et al., 2006*). The two *P. aeruginosa* strains could represent a parent strain that acquired resistance upon exposure to antibiotics, that wasn't fully cleared and so re-emerged; or a

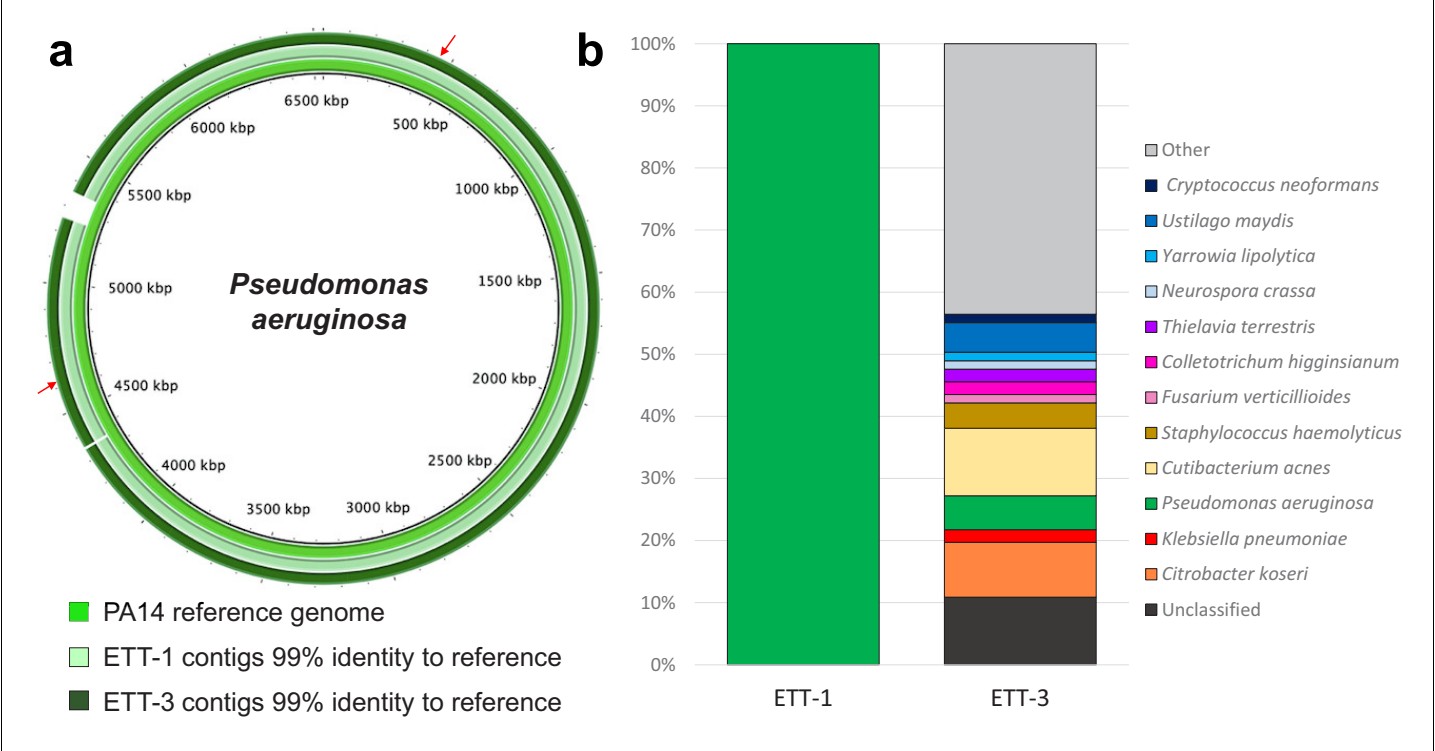

**Figure 2.** Direct metagenomic analysis of respiratory samples from a COVID-19 patient who developed a recurring VAP. (**a**) The sequence data were aligned to the genome of a *P. aeruginosa* reference strain PA14 (inner ring). The *P. aeruginosa* genomes from both ETT-1 (middle ring) and ETT-3 (outer ring) had 99% identify to the PA14 reference genome and greater than 99% to each other. The position of the AMR conferring SNPs in the *P. aeruginosa* from ETT-3 are indicated by red arrows. (**b**) Assembled contigs were taxonomically classified using Kraken two with % of total contigs assigned to each species shown for ETT-1 and ETT-3. Species with only one contig (0.68% of total contigs) were grouped as 'other'.

new infection or colonisation event with a closely related AMR strain circulating within the ICU. Unfortunately, we are unable to tell which of these scenario is more likely from these data.

Having recovered from both a viral and a recurring bacterial infection, we also sought to analyse the kinetics of the patient's innate and adaptive immune response across his time in the ICU (at days 23, 28, 38, 45, and 58 from testing positive for SARS-CoV2, *Figure 1*), as previous work has demonstrated striking immunological features associated with COVID-19 patients (*Mathew et al., 2020*). Robust activation of a broad range of immune cell subsets was revealed by flow cytometry, when compared to healthy controls. What was particularly striking was the level of activation and proliferation of CD4+, CD8+, and TCR-γδ T-cells, that appeared to wane between days 23–28 but were subsequently boosted after day 28, concomitant with the onset of the secondary bacterial infection (*Figure 3a–c* and *Figure 3e–g*, gating strategies in *Figure 3—figure supplement 1*). Similar levels of activation of T-cells has been reported for subsets of COVID-19 patients in other studies (*Mathew et al., 2020*). From day 28 a robust and steady increase in the activation and proliferation of conventional and TCR-γδ-T-cells was evident with approximately 20% and 40% of total CD4+ and CD8+ T-cells co-expressing the activation markers HLA-DR and CD38, respectively by day 58. A similar steady increase in activation levels, albeit at lower magnitudes, was observed for Natural Killer (NK) CD56^dim and CD56^bright cells (*Figure 3d and h*, gating strategies in *Figure 3—figure supplement 1*). Similar perturbations in the frequency and activation phenotypes of monocytes, blood monocyte-derived macrophages as well as neutrophils could be detected from day 28 onwards (*Figure 3—figure supplement 2a–e*). A robust IgG response to both SARS-CoV-2 and *P. aeruginosa* antigen was also detected in this patient (*Figure 3—figure supplement 3*).

Given the scale of the cellular response of this patient, we investigated whether the large expansions of CD4+ and CD8+ T-cells were due to a T-cell response targeting SARS-CoV-2, the secondary bacterial infection or to bystander T-cell activation (*Sandalova et al., 2010*; *Rivino et al., 2015*). We included the analysis of bystander activation, which is T-cell receptor-independent and cytokine-

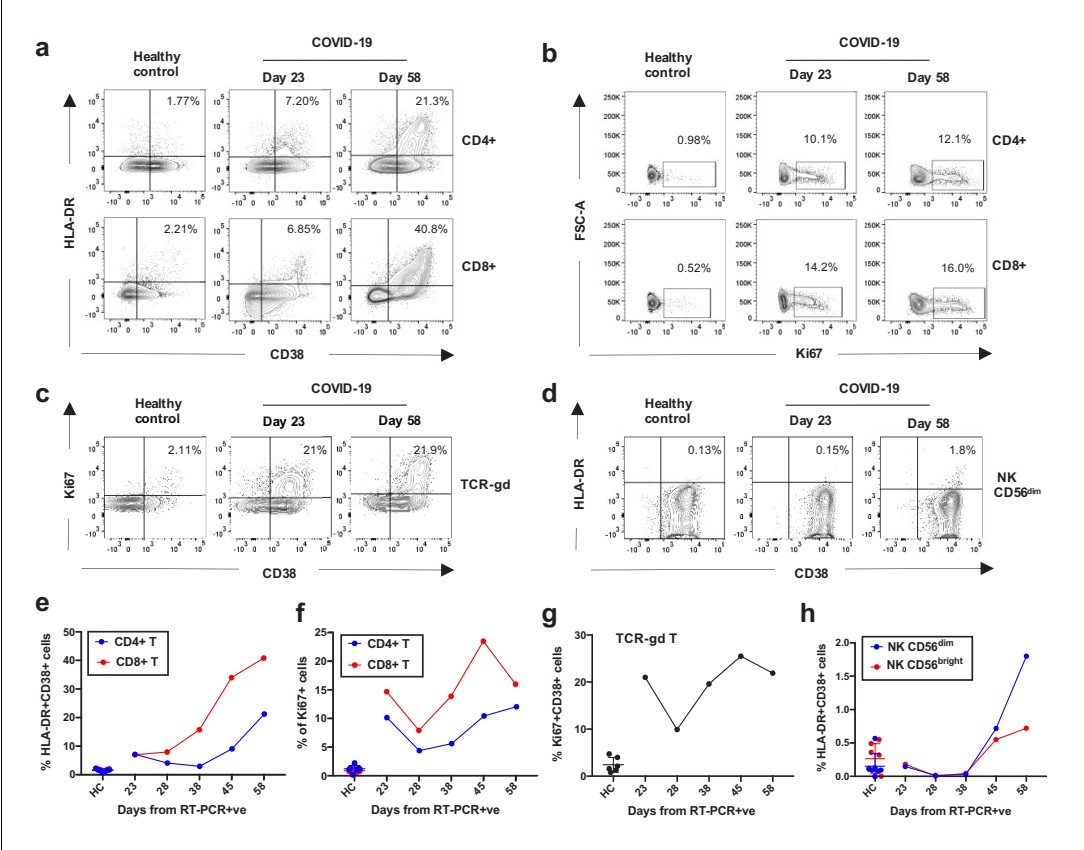

**Figure 3.** Kinetics of the adaptive and innate immune response in a COVID-19 patient during ICU treatment. (**a–d**) Representative flow cytometry plots showing the expression of activation and proliferation markers on CD4+ and CD8+ T cells (top and bottom panels, respectively in **a**, **b**), TCR-γδ T cells (**c**) and NK cells (**d**: plots shown for NK CD56$^{dim}$ cells) in a healthy control and the COVID-19 patient (at days 23 and 58). (**e-f**) Immune activation and proliferation levels are summarized for healthy controls (HC, n = 7) and five longitudinal time points of the COVID-19 patient and are assessed as co-expression of HLA-DR and CD38 (**e**) and expression of Ki67 (**f**) in CD4+ and CD8+ T cells, co-expression of CD38 and Ki67 by TCR-γδ T cells (**g**), and co-expression of HLA-DR and CD38 in NK CD56$^{dim}$ cells (**h**). Gating strategies for each population are included in *Figure 3—figure supplement 1*. All data is obtained by flow cytometry and samples were acquired on a Becton Dickinson LSR Fortessa X-20.

The online version of this article includes the following figure supplement(s) for figure 3:

**Figure supplement 1.** The gating strategies and flow cytometry plots are shown for each immune population.

**Figure supplement 2.** Kinetics of the innate immune response in the COVID-19 patient.

**Figure supplement 3.** Antibody response towards *Pseudomonas aeruginosa* and *SARS-CoV-2*.

mediated, as it is known to occur during other types of acute viral infection (*Sandalova et al., 2010*; *Rivino et al., 2015*). The role of this type of bystander activated T-cells in the recovery of patients remains largely unclear, as these cells can participate in protective immunity towards the virus but can also contribute to tissue damage (*Sandalova et al., 2010*; *Rivino et al., 2015*; *Kim and Shin, 2019*; *Maini et al., 2000*). We performed a brief stimulation of the patient's peripheral blood mono-nuclear cells (PBMCs), with or without overlapping peptides spanning the sequences of SARS-CoV-2 proteins (spike, membrane (M) and nucleoprotein (N)); of an immunodominant *P. aeruginosa* antigen (OprF); as well as an immunodominant human Cytomegalovirus protein (HCMV pp65) as an indica-tion of non-T-cell receptor (TCR) driven bystander activation. This was followed by intracellular cyto-kine staining to detect production of IFN-γ and TNF-α by the specific T cells. Following the encounter with specific peptides we observed a robust CD4+ T-cell response to SARS-CoV-2 which decreased from day 23 to day 58, and a more modest SARS-CoV-2-specific CD8+ T-cell response. This is in line with published work, suggesting higher magnitudes of SARS-CoV-2-specific CD4+ ver-sus CD8+ T-cells in severe COVID-19 patients (*Peng et al., 2020*), as well as consistent detection of virus-specific CD4+ T cells in recovered patients (*Grifoni et al., 2020*; *Figure 4a–c* and *Figure 4—*

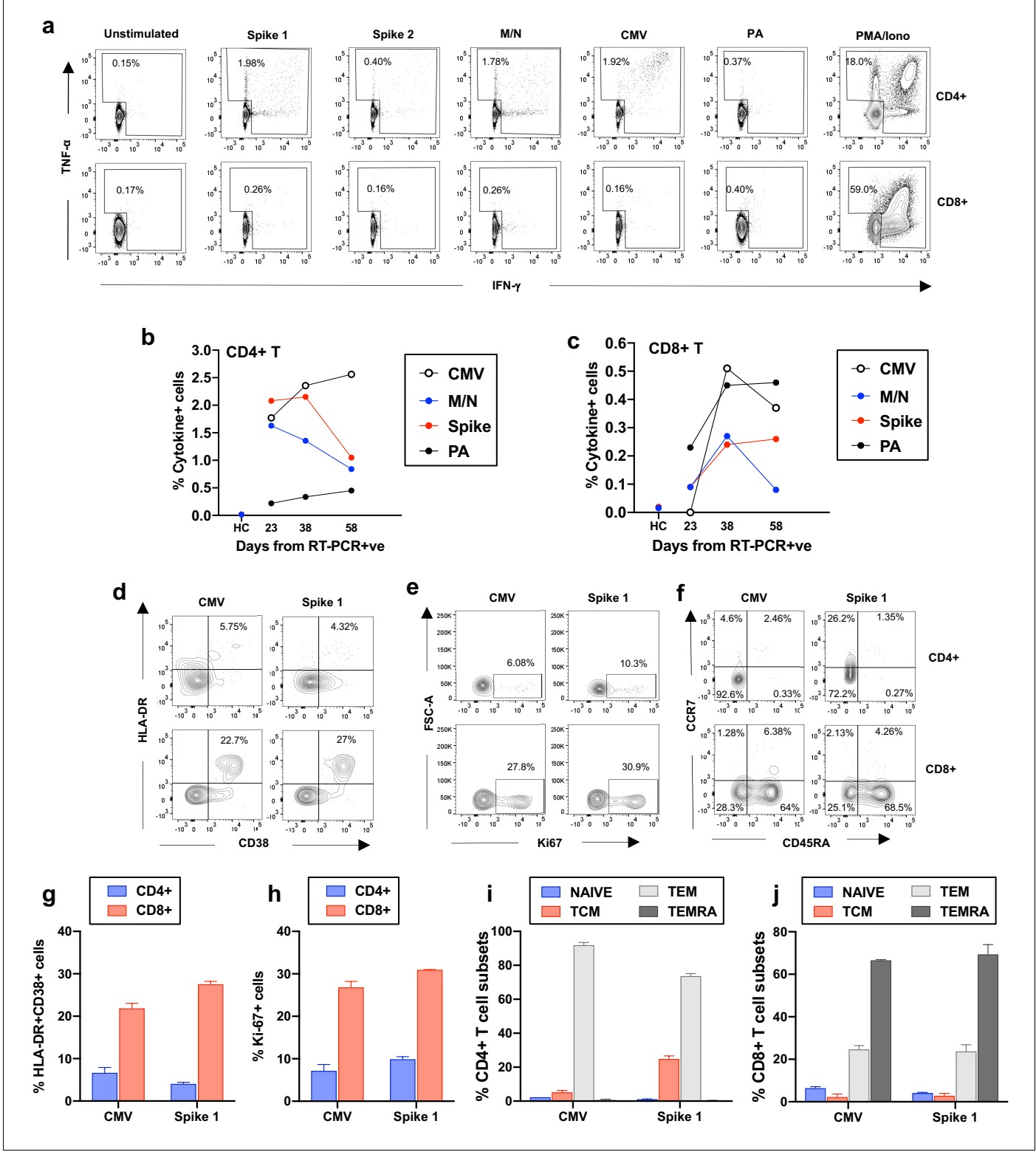

**Figure 4.** Frequency and phenotype of antigen-specific T cells. (**a**) Representative flow cytometry plots showing IFN-γ and TNF-α production by CD4+ and CD8+ T cells from the COVID-19 patient (day 23) as assessed by intracellular cytokine staining after a brief stimulation of PBMCs with HCMV pp65 (CMV), SARS-CoV-2 membrane, nucleoprotein (M/N respectively), spike 1, spike 2, and *Pseudomonas aeruginosa* OprF (PA) peptides. PMA/ionomycin and unstimulated conditions represent positive and negative controls. (**b-c**) Summarized are T-cell responses shown as % of IFN-γ and/or TNF-α+ cells in CD4+ or CD8+ T cells after subtraction of the negative controls in the COVID-19 patient at three longitudinal time points and in an age-matched

*Figure 4 continued on next page*

Figure 4 continued

healthy control (spike and M/N). (d-f) Representative flow cytometry plots showing T-cell expression of activation, proliferation and memory markers in CMV-specific and spike1-specific T cells in the COVID-19 patient (day 38), as assessed after a brief stimulation of PBMCs with CMV or spike 1 peptides followed by staining with antibodies targeting the relevant markers. Plots are gated on cytokine-producing cells (TNF-α and/or IFN-γ) in CD4+ or CD8+ T cells (top and bottom panels, respectively). (g-h) T-cell activation and proliferation assessed as co-expression of HLA-DR and CD38, and expression of Ki67 are shown for CD4+ and CD8+ CMV and spike 1-specific T cells (g and h, respectively). (i-j) Percentage of CMV and spike 1-specific T cells contained within each subset identified by expression of CCR7 and CD45RA is shown for CD4+ and CD8+ T cells (i and j, respectively). Naïve: CCR7 +CD45RA+; central memory (TCM): CCR7+CD45RA-; T effector memory (TEM):CCR7-CD45RA- and T effector memory RA re-expressing (TEMRA): CCR7-CD45RA+. Columns represent mean +/- SD of the experimental duplicates. All data is obtained by flow cytometry and samples were acquired on a Becton Dickinson LSR Fortessa X-20.
The online version of this article includes the following figure supplement(s) for figure 4:

**Figure supplement 1.** Experimental duplicates of intracellular cytokine staining data.

figure supplement 1a–b). CD4+ and CD8+ T-cell responses targeting *P. aeruginosa* peptides could also be detected, and increased over time concomitant with the onset of the recurring bacterial infection (*Figure 4b–c*).

The number of T-cells responding specifically to SARS-CoV-2 and *P. aeruginosa* represent only a fraction of those activated, which suggests that many of those detected in the patient's blood may be bystander activated. In support of this we detected an increased magnitude of CMV-specific T-cell responses over time, particularly in the CD8+ compartment (*Figure 3b–c*). We therefore considered whether CMV-specific T-cells were activated and proliferating in the blood of the COVID-19 patient. To test this we performed a brief stimulation of PBMCs with or without HCMV pp65 peptide (CMV) or SARS-CoV-2 spike 1 one peptide pools (the predominant SARS-CoV-2 target in this patient, within those tested) or with a positive control, and assessed the expression of the activation and proliferation markers HLA-DR, CD38 and Ki67 by the cytokine-producing cells that were responding to the peptide pools. Peptide stimulations and negative controls were performed in duplicate wells (*Figure 4—figure supplement 1a–b*). Both CMV and spike-1-specific CD8+ T-cells showed increased co-expression of HLA-DR and CD38 and of Ki67, indicating that these cells were activated and proliferating ex vivo, while this was less evident for their CD4+ T-cell counterparts (*Figure 4d–e and g–h*). This observation is in line with previous reports of bystander T-cell activation for CMV-specific CD8+ T-cell populations (*Sandalova et al., 2010*; *Rivino et al., 2015*). We furthermore analysed the phenotype of CMV and spike 1-specific T-cells on the basis of expression of CCR7 and CD45RA, which identifies naïve (CCR7+ CD45RA+), central memory (TCM: CCR7+ CD45RA-), effector memory (TEM: CCR7- CD45RA-) and effector memory re-expressing RA T-cells (TEMRA: CCR7-CD45RA+). CMV and spike 1-specific CD4+ T cells were mainly contained respectively within the TEM and the TEM/TCM subsets, while CMV and spike 1-specific CD8+ T cells were mainly contained respectively within the TEMRA and the TEM subsets. The predominant TEMRA phenotype of SARS-CoV-2 specific CD8+ T-cells in convalescent COVID patients is in line with recent work (*Neidleman et al., 2020*). The presence of a large proportion of CD8+ TEMRA within the CMV and spike 1-specific CD8+ T cell populations also supports the notion that these cells are stimulated through cytokine receptors rather than through the TCR, as TEMRA cells are believed to arise following antigen withdrawal and cytokine-mediated stimulation (for e.g. IL-15), as TCR ligation downregulates CD45RA expression both in vitro and during responses in vivo (*Henson et al., 2012*).

In summary, the immune profiles show robust innate and adaptive immune activation in a COVID-19 patient experiencing a secondary bacterial infection, and a detectable T-cell and IgG response targeting both SARS-CoV-2 and *P. aeruginosa*. Furthermore, we show that CD8+ T-cells targeting viral antigens from SARS-CoV-2 and from the persistent virus HCMV display expression of markers of activation and proliferation and are predominantly TEMRA and TEM cells. Although it is a challenging task to demonstrate that these cells have not encountered their cognate viral antigens in vivo, the TEMRA phenotype of these cells makes it unlikely and suggests that these cells may be proliferating in a bystander way through the action of cytokines present in the inflammatory milieu.

## Discussion

In this case report we demonstrate the potential of the application of multidisciplinary technologies to longitudinally-collected patient samples to define the complex dynamics of patient-pathogen-therapy interactions in real-time. Metagenomics directly applied to respiratory samples facilitated the identification of the SNPs responsible for the AMR phenotype of the later *P. aeruginosa* isolate. However, the most striking feature of this COVID-19 case was the escalating number of circulating activated T-cells more than two months after testing positive for SARS-CoV-2, and 6 weeks after clearing the viral infection. While we could attribute some of this to the recurring bacterial infection, the scale of the activation considered alongside our evidence of increased frequencies of T-cells specific for unrelated antigens, suggests there may be a significant amount of bystander activation.

Studies on large COVID-19 patient cohorts have shown that the period of the peak of T-cell activation in COVID-19 is prolonged compared to other acute viral infections or after vaccination with live attenuated viruses (*Mathew et al., 2020*), which may be due to the inability of the immune system to downregulate its response. Here, we speculate that due to SARS-CoV2 infection the patient displays a heightened immune system, which was further stimulated by the recurring *P. aeruginosa* infections, which led to bystander activation of T cells specific for antigens unrelated to either SARS-CoV2 or *P. aeruginosa*. Bystander T-cell activation could have played a critical role in the severity of illness and the longer-term complications associated with the development of post-acute COVID-19 experienced by this patient (*Grifoni et al., 2020*). Given the recent appreciation for the role of corti-costeroids in reducing the risk of death following SARS-CoV-2 infection by 20% (*Sterne et al., 2020*), this case suggests that targeting its use to patients with immunological responses as described here, may also improve their long-term recovery.

## Materials and methods

### Patient recruitment

The patient was enrolled onto the DISCOVER study (Diagnostic and Severity markers of COVID-19 to Enable Rapid triage study), a single centre prospective study recruiting consecutive patients admitted with COVID-19, from 30.03.2020 until present (Ethics approval via South Yorkshire REC: 20/YH/0121, CRN approval no: 45469). Blood/serum samples from pre-pandemic healthy controls and asymptomatic healthy controls were obtained under the Bristol Biobank (NHS Research Ethics Committee approval ref 14/WA/1253).

### Clinical microbiology

All ETT samples were processed in the Severn Infection Services laboratory as per standard operating procedures. Antibiotic susceptibility testing was performed by either disk diffusion (according to EUCAST version nine guidance *EUCAST, 2019*) or on a VITEK two machine (BioMeriux, France).

### SARS-CoV-2 test

SARS-CoV-2 test was performed by an in-house RT-PCR at the regional South West Public Health England Regional Virology laboratory, utilising a PHE approved assay at the time of testing.

### DNA extractions

DNA from the endotrachaeal aspirates were extracted using the CTAB/Phenol:Chloroform:Isoamyl alcohol and bead-beating approach of *Griffiths et al., 2000* with modifications of *DeAngelis et al., 2009*. Phase lock gel tubes (ThermoFisher) and linear polyacrylamide (Sigma) were included to increase nucleic acid yields and total DNA was resuspended in 50 µl of DNase/RNase free water before storage at −20℃.

### DNA sequencing

Samples were prepared for sequencing using SQK-LSK109 kit (Oxford Nanopore) with 1 µg DNA starting input as per manufacture's protocol. Briefly 1 µg of DNA was end repaired and a tailed using NEBNext Ultra II module E7546 (3.5 µl End Repair Buffer, 2 ul FFPE repair mix, 3.5 µl Ultra II end-prep reaction buffer and 3 µl of Ultra II end-prep enzyme mix to 1 µg DNA in a total of reaction

volume of 60 μl). This was incubated at 20°C for 5 min followed by 65°C for 5 min. Clean-up was performed using AMPure XP beads (Beckman Coulter) in a 1X ratio. Adaptors were ligated by adding 5 μl adaptor mix (Oxford Nanopore) 25 μl ligation buffer (Oxford Nanopore) and 10 μl Quick T4 ligase (NEB Module MO202). Following a 20 min incubation at room temperature the adaptor ligated DNA was cleaned using AMPure beads in a 0.8 X ratio and washed using Long Fragment Buffer (Oxford Nanopore) before eluting in 25 μl of elution buffer (Oxford Nanopore). Final quantification by fluorometry (Qubit) was performed and 300 ng DNA prepared for sequencing according to the manufacturer's instructions (Oxford Nanopore). Sequencing was performed on a PromethION R9.4.1 flow cell (FLO-PRO002) and run for 48 hr using live basecalling, files were outputted in Fast5 and Fastq format. Human-filtered sequencing data for this study have been deposited in the European Nucleotide Archive (ENA) at EMBL-EBI under accession PRJEB40239.

## Bioinformatics

Bioinformatics was orchestrated using reticulatus (https://github.com/SamStudio8/reticulatus/ *Köster and Rahmann, 2012*) with configuration: Flye (v2.6) metagenomic assembly (*Kolmogorov et al., 2019*), Racon polishing (v1.4.9 + GPU, two rounds) (*Vaser et al., 2017*) and Medaka polishing (v0.8.0 + GPU, one round). Contigs were assigned to a taxon by Kraken 2 (*Wood et al., 2019*). SNP differences between the *P. aeruginosa* contigs for ETT-1 and ETT-3 was determined using *Mauve* (*Darling et al., 2004*). Contigs were loaded into BRIG (v0.95) (*Alikhan et al., 2011*) as concentric rings and compared against the PA14 or JCSC1435 reference genomes using blastn (ncbi-blast 2.10.1+) (*Alikhan et al., 2011*). MLST types were identified by mapping contigs against *P. aeruginosa* or *S. haemolyticus* according to the schemes held in the pubmlst databases (*Arnold et al., 2020*).

## PBMC isolation

Blood samples were collected from the COVID-19 patient in EDTA vacutainer tubes and PBMCs isolated from peripheral blood by Ficoll gradient purification and cryopreserved. Healthy donor PBMCs were obtained from Bristol Biobank (REC: 14/WA/1253).

## Synthetic peptides

15-mer peptides overlapping by 10 amino acids and spanning the sequences of SARS-CoV-2 spike (Accession Number: NC_045512.2, Protein ID: YP_009724390.1) and HCMV pp65 (AD169 strain) were purchased from Mimotopes (Australia). The purity of the peptides was >80% (Spike) or >70% (pp65) and peptides were dissolved as described previously (*Rivino et al., 2015*). SARS-CoV-2 M and N Peptivator peptide pools were purchased from Miltenyi. An OprF (PA1777) peptide library comprising 20-mer peptides overlapping by 10 amino acids was synthesized by GL Biochem Ltd., Shanghai, China (*Quigley et al., 2015*).

## Flow cytometry staining and PBMC stimulation

PBMCs were thawed and either stained ex vivo or stimulated in AIMV 2% FCS with or without peptide pools from SARS-CoV2 spike, M, N, HCMV pp65 (all 1 μg/ml), OrpF PA (10 μg/ml) or with PMA/iono (PMA 10 ng/ml, Iono 100 ng/ml, Sigma Aldrich) for 5 hr at 37°C in the presence of brefeldin A (BD, 5 μg/ml). To assess degranulation, CD107a FITC antibody was added to the cells at the beginning of the stimulation. Cells were stained with a viability dye Zombie Aqua (Biolegend) for 10 min at room temperature and with antibodies targeting surface markers (20 min 4°C, diluted in PBS (HyClone) 1% BSA (Sigma Aldrich)). Cells were fixed for 45 min/overnight in eBioscience Foxp3/Transcription factor fixation/permeabilization buffer (Invitrogen) and intracellular staining was performed using eBioscience Foxp3/Transcription factor permeabilization buffer (Invitrogen) for Ki67 or intracellular cytokines (30 min on ice). Cells were acquired on a BD Fortessa X20 and data analysed using FlowJo software v10.7. A complete list of antibodies is included in *Supplementary file 1*.

## Antibody quantification

a. *Pseudomonas Aeruginosa:* The PA antigen OprF was diluted in coating buffer, added to the wells of MaxiSorp flat-bottom 96-Well immunoplates (Thermo Fisher Scientific, USA) and incubated for 2 hr at 37°C. Immunoplates were blocked with 1% BSA/PBS for 1 hr at 37°C. Sera from each sample

were diluted and added to the immunoplates in triplicate and incubated at 4°C overnight. Biotin mouse anti-human IgG (BD Pharmingen, USA) was added to the plate and incubated for 1 hr at RT. Streptavidin-HRP (R and D Systems, UK) was added and incubated for 30 mins at RT. ELISAs were developed using TMB substrate. Results were recorded using a Multiskan GO Microplate Spectrophotometer (Thermo Scientific, USA) and absorbance values (OD 450) were converted to ELISA units. OprF antigen was produced using the recombinant vector pSUMO-OprF as described previously (*Quigley et al., 2015*). b. ELISA methods were based on the protocols for measuring SARS-CoV-2 seroconversion (*Amanat et al., 2020*; *Stadlbauer et al., 2020*) with some modifications. A full-length trimeric stabilized version of the SARS-CoV-2 Spike protein was expressed in insect cells using the MultiBac baculovirus expression system and affinity purified, whereas the SARS-CoV-2 N protein was expressed in, and purified from, *E. coli*. High-binding plates (MaxiSorp, NUNC) were coated with either 10 mg/ml Spike or 20 mg/ml N protein in PBS and left overnight at 4°C followed by blocking for 1 hr (3% BSA/0.1% Tween/PBS). Dilution buffer (1% BSA/0.1% Tween/PBS) containing patient plasma samples collected at different time points, or serum controls, were added to wells either in duplicate (for patient, pre-pandemic (PP) or pooled serum standard (PS)) or in single wells (healthy controls, HC) at a final dilution of 1 in 450 (adjusting for dilution of plasma if obtained via PBMC collection) and left for 2 hr at RT. Goat anti-human IgG-HRP (Southern Biotech) was added at 1 in 25,000 in dilution buffer for 1 hr, followed by development with OPD solution (Sigma) as per the manufacturer's instructions. Development was quenched after 30 min using 3M HCl and absorbance measured at 492 and 620 nm using a BMG FLUOstar Omega Spectrophotometer. Plots show mean and standard deviation where samples were run in duplicate.

Control samples: Serum from n = 6 adult pre-pandemic donors recruited in 2008 for a vaccine trial were obtained from the Bristol Biobank (NHS REC 14/WA/1253) and were used as known-negative controls on the same plate as the COVID19 case samples, in duplicate. In addition, we have included the OD values from the healthy control donor samples which were used as controls for the T cell assays; these donors were collected during the pandemic but reporting no exposure or symptoms prior to sampling (also obtained via the Biobank as explained in the manuscript methods section). These samples were run in single wells on different plates to the COVID19 case samples; however, all plates included a pooled serum (PS) standard control, in duplicate, to assess for interplate variability. The PS average ODs across six plates are therefore shown. The PS, HC and PP samples used in this experiment were heat inactivated at 56°C for 30 min, while the patient plasma were not (after testing negative for SARS-CoV-2). Despite the difference in sample type and handling, we are confident these do not impact on antibody levels or ELISA signal from our own in-house comparisons (data not shown) as well as data from others (*Amanat et al., 2020*).

## Acknowledgements

We would like to thank Kapil Gupta and Imre Berger for kindly providing us the spike protein, Natalie Di Bartolo and Ashley Toye for kindly providing us the N protein, both used for the SARS-CoV-2 serology work. The authors wish to acknowledge the assistance of Dr. Andrew Herman and Helen Rice and the University of Bristol Faculty of Biomedical Sciences Flow Cytometry Facility. We would also like to thank Keith Jolley and the Bristol University UNCOVER team for helpful discussions during the execution of this work and preparation of the manuscript. This work was supported by donations to Southmead Hospital Charity (Registered Charity Number: 1055900), by the Wellcome Trust (reference number: 212258/Z/18/Z) and by the Elizabeth Blackwell Institute, University of Bristol, with funding from the University's alumni and friends. DKB is supported by a Cystic Fibrosis Trust PhD studentship (CF Trust SRC 015). RJB and DMA are supported by UKRI (MR/S019553/1 and MR/R02622X/1)

## Additional information

### Funding

| Funder | Grant reference number | Author |
| --- | --- | --- |
| Southmead Hospital Charity | | Fergus Hamilton |

| Wellcome Trust | 212258/Z/18/Z | Ruth C Massey |
| Elizabeth Blackwell Institute | | Laura Rivino |
| UK Research and Innovation | MR/S019553/1 | Rosemary J Boyton |
| UK Research and Innovation | MR/R02622X/1 | Daniel M Altmann |
| Cystic Fibrosis Trust | CF Trust SRC 015 | David K Butler |

The funders had no role in study design, data collection and interpretation, or the decision to submit the work for publication.

## Author contributions
Michaela Gregorova, Ruth C Massey, Conceptualization, Resources, Data curation, Formal analysis, Supervision, Funding acquisition, Validation, Investigation, Visualization, Methodology, Writing - original draft, Project administration, Writing - review and editing; Daniel Morse, Tarcisio Brignoli, Data curation, Formal analysis, Investigation; Joseph Steventon, Sam Nicholls, Data curation, Formal analysis, Investigation, Methodology; Fergus Hamilton, Conceptualization, Resources, Methodology, Writing - review and editing; Mahableshwar Albur, David Arnold, Matthew Thomas, Conceptualization, Resources, Writing - review and editing; Alice Halliday, Conceptualization, Resources, Formal analysis, Investigation, Methodology; Holly Baum, Christopher Rice, Elizabeth Oliver, Claire McMurray, Formal analysis, Investigation, Methodology; Matthew B Avison, Formal analysis, Investigation, Methodology, Writing - review and editing; Andrew D Davidson, Marianna Santopaolo, Formal analysis, Investigation, Writing - review and editing; Anu Goenka, Resources, Investigation, Methodology, Writing - review and editing; Adam Finn, Resources, Supervision, Writing - review and editing; Linda Wooldridge, Resources, Supervision, Funding acquisition, Writing - review and editing; Borko Amulic, Rosemary J Boyton, Daniel M Altmann, Resources, Formal analysis, Supervision, Funding acquisition, Writing - review and editing; David K Butler, Resources, Formal analysis, Supervision, Funding acquisition, Investigation, Writing - review and editing; Joanna Stockton, Investigation, Methodology; Charles Cooper, Data curation, Formal analysis, Visualization; Nicholas Loman, Conceptualization, Resources, Formal analysis, Supervision, Visualization; Michael J Cox, Conceptualization, Resources, Supervision, Funding acquisition, Project administration, Writing - review and editing; Laura Rivino, Conceptualization, Resources, Data curation, Formal analysis, Supervision, Funding acquisition, Investigation, Methodology, Writing - original draft, Project administration, Writing - review and editing

## Author ORCIDs
Michaela Gregorova (iD) http://orcid.org/0000-0003-1605-0558
Mahableshwar Albur (iD) http://orcid.org/0000-0001-9792-7280
David Arnold (iD) http://orcid.org/0000-0003-3158-7740
Andrew D Davidson (iD) http://orcid.org/0000-0002-1136-4008
Laura Rivino (iD) https://orcid.org/0000-0001-6213-9794
Ruth C Massey (iD) https://orcid.org/0000-0002-8154-4039

## Ethics
Human subjects: The patient was enrolled onto the DISCOVER study (Diagnostic and Severity markers of COVID-19 to Enable Rapid triage study), a single centre prospective study recruiting consecutive patients admitted with COVID-19, from 30.03.2020 until present (Ethics approval via South Yorkshire REC: 20/YH/0121, CRN approval no: 45469). Blood/serum samples from pre-pandemic healthy controls and asymptomatic healthy controls were obtained under the Bristol Biobank (NHS Research Ethics Committee approval ref 14/WA/1253).

## Decision letter and Author response
Decision letter https://doi.org/10.7554/eLife.63430.sa1
Author response https://doi.org/10.7554/eLife.63430.sa2

## Additional files

### Supplementary files

- Supplementary file 1. Details of the antibodies used for the flow cytometry experiments.

- Transparent reporting form

### Data availability

Human-filtered sequencing data for this study have been deposited in the European Nucleotide Archive (ENA) at EMBL-EBI under accession PRJEB40239.

The following dataset was generated:

| Author(s) | Year | Dataset title | Dataset URL | Database and Identifier |
|---|---|---|---|---|
| Gregorova M | 2020 | Metagenomic analysis of respiratory samples from a COVID-19 ICU patient | https://www.ebi.ac.uk/ena/browser/text-search?query=PRJEB40239 | European Nucleotide Archive, PRJEB40239 |

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
