## [Decision Letter]

**Acceptance summary:**

The article demonstrates the capacity of multidisciplinary technologies to provide illustrative insights into hitherto hidden aspects of the evolution of immune responses during a complicated SARS-CoV2 infection and recovery. In particular, the prominent bystander T cell activation in hospital and escalating number of circulating activated T-cells more than two months after testing positive for SARS-CoV-2, and six weeks after clearing the viral infection are notable. It is unusual to get this level of detail, albeit for a single patient, and hopefully this would lead to further thinking and research about how to approach and treat patients with long COVID.

**Decision letter after peer review:**

[Editors’ note: the authors submitted for reconsideration following the decision after peer review. What follows is the decision letter after the first round of review.]

Thank you for choosing to send your work, "Post-acute COVID-19 associated with evidence of bystander T-cell activation and a recurring AMR bacterial pneumonia", for consideration at *eLife*. Your submission has been assessed by a Senior Editor in consultation with a member of the Board of Reviewing Editors. Although the work is of interest, we regret to inform you that the findings at this stage are too preliminary for further consideration at *eLife*.

Specifically, the major conclusion, namely bystander T cell activation in severe COVID19, even though plausible, is not justified from the data presented in the manuscript. Even with the short form assessment, the manuscript would require significant amount of additional data from multiple patients of different severity and co-infection patterns to make any claims on the finding of bystander activation.

Reviewer #1:

Gregarova et al., present a clinical case of COVID-19 and ventilator acquired *P. aeruginosa* co-infection with persistent T cell activation, mostly of a bystander nature. While the *Pseudomonasaeruginosa* strain was initially found to be antibiotic sensitive and treated accordingly, within eight days the bacterial infection re-emerged. Whole genome sequencing revealed a mixed infection of *P. aeruginosa* associated with AMR infections and a coagulase negative *S. haemolyticus* strain. The authors profiled activation status of monocytes and T cells during the course of the hospitalization of the patient. By profiling antigen specific responses, they identified that a large proportion of activated T cells that neither recognized SARS-CoV2 nor *P. aeruginosa* antigens. Responses to HCMV were evident in the patient. Such bystander T cell responses are known in other viral infections also, although the relevance of these is not well understood.

1) This study uses elegant molecular tools to quantify T cell responses in the patient. The finding that there is an increase in activated bystander T cells during the course of the disease may be important. The authors make an important point about the relevance of this finding to potential use of corticosteroids in treatment of severe COVID19. However, this study does not have the power to inform whether ventilator acquired co-infections are the drivers of this bystander activated state in COVID19 patients. A larger study with defined groups of SARS-CoV2 with and without VAP infection, and VAP infections without SARS-CoV2 infection would be required to evaluate the relevance of the bystander T cell response in COVID19 disease severity/outcome.

Reviewer #2:

The manuscript by Gregorova et al., investigates a COVID-19 patient with an antibiotic-resistant secondary bacterial infection by *Pseudomonas aeruginosa* evident with bystander T cell activation. The case study demonstrates the potential application of multidisciplinary technologies to longitudinally collected patient samples to define the complex dynamics of patient-pathogen-therapy interactions in real-time. Though the importance and novelty of the timely research, the lack of appropriate sample size in the key experiment limits the overall significance of this work.

1) This is a practical matter, but a significant one. How confident are the authors to claim any statement from a single patient case study? Some of the immune cell response data look dramatic. However, with only one data point, no biological replicates and no possible statistics is not very convincing.

2) It's unclear how the authors chose and prepared the controls, the "pre-pandemic healthy controls" and "asymptomatic healthy controls" are not well explained. And this part of immune assay is not clearly documented while this is critical, like are they undergone the same treatment, and what are the vital statistics?

3) It's not clear how the presence of *P. aeruginosa* is associated with the disease severity or pneumonia. One of such confusing statement is "There were no further concerns about a bacterial infection during his ICU stay although a third *P. aeruginosa* was cultured from a sample (ETT- 3) collected two days after he completed his course of meropenem, that was resistant to both pip/taz and meropenem (MICs >16 mg/l and >8mg/l respectively)" in the Introduction. Does this mean in ETT-3, this bacterium is still detected and resistant to antibiotics? But why it becomes not a concern? If it's because the patient is in his remission stage, it makes people wonder if the earlier concern from the bacterial infection during ETT-1 and ETT-2 is a side effect from the severe SARS-CoV2 symptoms instead of any causation or association effects.

4) How's the level of *P. aeruginosa* after the patient is discharged from hospital and even before the development of SARS-CoV-2? Although it's too late to ask such a question about the level before SARS-CoV-2, is it possible that *P. aeruginosa* is an existing bacterium in this patient other than from a secondary bacterial infection after SARS-CoV-2?

5) What is the logic of performing sequencing on ETT-1 and ETT-3 but not ETT-2? ETT-2 genome sequencing might help to show some evidence of evolution of the mutation points.

6) During metagenomics, did the authors see other microbes? It's surprising if there are only two bacterial species detected in those samples using metagenomic sequencing. Or is it because these two strains are the most dominant ones that detected? If so, it would be useful to show the whole microbial community profile from this sequencing.

7) Since there are two strains that are fully sequenced, although *Staphylococcus haemolyticus* is "not recognized as lung pathogens" as stated by the authors, if this particular strain is resistant to methicillin, would it also be a potential target strain with AMR deteriorating the disease in this scenario? Did the authors also detect it from ETT-3? Would it be necessary for the authors to carry out a similar assay as Figure 3E and f using the *S. haemolyticus* antigen to show if it has an effect?

8) It's very interesting observation that CD4^+^, CD8^+^ and TCR-gd T-cells appear to wane between days 23-28 and boosted after day 28, is there an antibiotic effect here as pip/taz was administrated ~day 28? How to explain a decrease around day 45 on Figure 3B? Was that also an antibiotic effect by the starting of meropenem? It could be further discussed in the paper. And authors should explain a little more in the text about the difference between Figure 3A and 3B.

Reviewer #3:

Dr Gregorova and colleagues present an interesting case report entitled "Post-acute COVID-19 associated with evidence of bystander T-cell activation and a recurring antibiotic-resistant bacterial pneumonia." The report details a patient that presented with respiratory failure 20 days after testing positive for COVID-19 on screening following exposure to a family member. The patient's course was complicated by two episodes of VAP with metagenomics serially revealing complete genomes of *P. aeruginosa* encoding antibiotic resistance. The fundamental thrust of the paper is the inference that both the initial respiratory failure and the subsequent VAP episodes result directly or indirectly from immunopathology driven by bystander T cell activation, which the investigators assess by determining CD4 and CD8 cell cytokine expression following PBMCs exposure to viral and bacterial peptides.

The report is easy to read and proposes an explanation for the patient's respiratory failure that is congruent with known immune responses to SARS-CoV-2. Indeed, as the authors note, this may help explain the observed efficacy of dexamethasone in patients with respiratory insufficiency. However, there are limitations that reduce enthusiasm.

1) The paper broadly suggests/implies that bystander activation of T cells is a proximate cause of the VAP episodes without any direct evidence to support this notion. It is well established in the literature that post-viral leukocyte dysfunction (most often lymphocytes and monocyte/macrophage lineage cells) enhance susceptibility to secondary bacterial infections. In this case, the authors demonstrate variably elevated numbers of T cells but do not demonstrate failures of *P. aeruginosa* sensing/responding/clearance by any cell population.

2) The essential arguments for bystander activation are the stimulation experiments in Figure 3E-F. These data are widely variable and are difficult to interpret in the absence of data from the times that VAP was first suspected. The only shown dates are from ICU admission and two dates each after several days of VAP therapy.

3) In a single case report, it is always challenging to confidently know what is driving an observed immune response. This is particularly so in a patient who was well at the time of viral diagnosis but presents 20 days later. Certainly, the authors' inference that the respiratory failure represents COVID-19 immunopathology is plausible, even likely. However, since the interval between diagnosis and respiratory failure is longer than the mean interval reported in recent large series and since longer intervals provide more opportunities for patients to acquire new pathogens/be exposed to new insults, more robust evidence than a single case would typically be expected to make such a claim.

4) The metagenomics data are impressively generated, but don't drive new understanding. That an intubated patient might acquire two related strains of *P. aeruginosa* during a single episode of critical illness is unsurprising. As above, there seems to be an underlying implication that the T cell activation drives the SNP differences between the isolates, but that is not further demonstrated. Metagenomic assessment at the time of ICU admission might have been supportive of the authors' assessment that there was no active viral infection (SARS-CoV-2 or otherwise) driving the acute decompensation that led to intubation.

[Editors’ note: further revisions were suggested prior to acceptance, as described below.]

Thank you for your note appealing the decision on your article "Post-acute COVID-19 associated with evidence of bystander T-cell activation and a recurring AMR bacterial pneumonia" for reconsideration by *eLife*.

We would like to assure you that the single case did not underscore rejection. We do consider case reports if they change the practice of medicine, bring to the fore important clinical questions and/or trigger new patient-centered research. Your report fell into the first category. With that said, there were significant concerns that I have listed below in consultation with the Reviewing Editor who handled your paper, and we would be happy to send it out for re-review, without a guarantee of acceptance, once we receive a point-to-point response to the critique in the reviews that were sent to you previously.

Specifically, we would consider a rebuttal that focuses on the comments of reviewer 3 and broadly answers the summarised concerns listed below. This may be done through additional clinical data and experimental data that the authors find feasible.

The case study's value is in suggesting that bystander T cell activation in SARS-CoV2 infection may have led to a severe COVID-19 presentation. This supports the benefit of corticosteroids in COVID-19. The message is indeed plausible and important. The authors need to tell us how much it adds to what we know (see Mathew et al., 2020).

The essence of the case report seems to be that a patient with asymptomatic SARS CoV2 infection developed respiratory symptoms suggestive of COVID19 at 2 weeks post-infection, requiring hospitalisation. After more than 96 hrs of hospitalisation the patient required ICU admission and was SARS CoV2 negative, but resistant P. Aeurginosa positive on ETT aspirate. At this point bystander T cell activation was noted. The authors link the bystander activation to the SARS-CoV2 infection and disease progression, which is plausible but uncertain due to questions: namely (a) whether an additional infection may have occurred in 2 weeks, (b) whether bystander activation measured at 3 weeks is confidently determined and relatable to original infection, and (c) whether the hospital-acquired multi drug resistant pneumonia is related to the immune dysfunction.

The specific concerns that need to be considered and addressed are:

a) Is this severe COVID-19 or a new insult given that 15 days passed from PCR positivity for onset of symptoms? A clear comparison of this to reported intervals between onset of infection and clinical deterioration would help make the case stronger. If this is very unusual, then alternative explanations, such as a new infection, become more likely than severe COVID-19. Clinical data would be helpful, including any repeat PCR at admission.

b) *P. aeruginosa* infection is probably hospital-acquired and multiple strains are quite possible. The link to bystander cell activation only arises if it led to ineffectual sensing/clearance. Can this be supported by any experimental evidence beyond clinical deterioration?

c) The essential arguments for bystander activation are illustrated in Figure 3E-F – that were found not to be particularly convincing. At the very least, please provide full data and experimental replicates.

---

## [Author Response]

[Editors’ note: The authors appealed the original decision. What follows is the authors’ response to the first round of review.]

Reviewer #1:Gregarova et al., present a clinical case of COVID-19 and ventilator acquired *P. aeruginosa* co-infection with persistent T cell activation, mostly of a bystander nature. While the Pseudomonas aeruginosa strain was initially found to be antibiotic sensitive and treated accordingly, within eight days the bacterial infection re-emerged. Whole genome sequencing revealed a mixed infection of *P. aeruginosa* associated with AMR infections and a coagulase negative S. haemolyticus strain. The authors profiled activation status of monocytes and T cells during the course of the hospitalization of the patient. By profiling antigen specific responses, they identified that a large proportion of activated T cells that neither recognized SARS-CoV2 nor *P. aeruginosa* antigens. Responses to HCMV were evident in the patient. Such bystander T cell responses are known in other viral infections also, although the relevance of these is not well understood.1) This study uses elegant molecular tools to quantify T cell responses in the patient. The finding that there is an increase in activated bystander T cells during the course of the disease may be important. The authors make an important point about the relevance of this finding to potential use of corticosteroids in treatment of severe COVID19. However, this study does not have the power to inform whether ventilator acquired co-infections are the drivers of this bystander activated state in COVID19 patients. A larger study with defined groups of SARS-CoV2 with and without VAP infection, and VAP infections without SARS-CoV2 infection would be required to evaluate the relevance of the bystander T cell response in COVID19 disease severity/outcome.

We are in absolute agreement with this reviewer on this matter. As an entirely descriptive case study of a single patient, and we make no statements within the manuscript as to there being a causal link between the VAP and the bystander activation observed. However, if the reviewer feels this has been inadvertently inferred anywhere within the manuscript, we would be more than happy to address this.

As discussed above, the editor has confirmed that case studies are considered suitable for publication in *eLife* if they “change the practice of medicine, bring to the fore important clinical questions and/or trigger new patient-centred research”. With this in mind we hope the reviewer will find themselves in agreement with us that our study satisfies this remit in that, just as they suggest, a study of a wider population of patients is warranted based on these findings.

Reviewer #2:The manuscript by Gregorova et al., investigates a COVID-19 patient with an antibiotic-resistant secondary bacterial infection by *Pseudomonas aeruginosa* evident with bystander T cell activation. The case study demonstrates the potential application of multidisciplinary technologies to longitudinally collected patient samples to define the complex dynamics of patient-pathogen-therapy interactions in real-time. Though the importance and novelty of the timely research, the lack of appropriate sample size in the key experiment limits the overall significance of this work.

This is an entirely descriptive case study of a single patient. As discussed above, the editor has confirmed that these types of studies are considered suitable for publication in *eLife* if they “change the practice of medicine, bring to the fore important clinical questions and/or trigger new patient-centred research”. With this in mind we hope the reviewer will find themselves in agreement with us that our study satisfies this remit in that it is an interesting study that raises important clinical questions and warrants further investigation in a wider cohort of patients. We will address the specific issue raised with regards to replicates below.

1) This is a practical matter, but a significant one. How confident are the authors to claim any statement from a single patient case study? Some of the immune cell response data look dramatic. However, with only one data point, no biological replicates and no possible statistics is not very convincing.

We are in complete agreement with the reviewers that no statement can be made with any confidence based on the findings from a single patient, which is why we were careful not to make any. We have re-read the manuscript to ensure this remains the case, however if the reviewer can point us to any, we may have inadvertently written we will happily temper them.

The issue with regards to biological replicates in a case study of a severely ill patient is complex. Biological replicates for larger immunological studies would typically include samples from multiple patients, which is beyond the remit of a case study as we were limited by the numbers of samples we could collect for research purposes. However, with what we were able to obtain, our immunophenotyping was performed on PBMCs from five longitudinal time points (Figure 3E-F) while intracellular cytokine staining was performed on 3 time points (Figure 4B-C) due to limiting cell numbers. For this revised manuscript we now provide technical replicates for the experiments described in the original manuscript consistent (Figure 4—figure supplement 1B). We also provide new data (in duplicate) strengthening the case for the bystander activation we have observed for this patient (Figure 4D-J). We hope this is sufficient to alleviate the reviewers concerns on this matter.

2) It's unclear how the authors chose and prepared the controls, the "pre-pandemic healthy controls" and "asymptomatic healthy controls" are not well explained. And this part of immune assay is not clearly documented while this is critical, like are they undergone the same treatment, and what are the vital statistics?

We thank the reviewer for pointing out this omission in the original manuscript. Details on the pre-pandemic and asymptomatic healthy controls and a more detailed description of the methodology for the serology assay have been included (Materials and methods section). These are copied below for your reference:

“ELISA methods were based on the protocols for measuring SARS-CoV-2 seroconversion outlined in Amanat et al., (2020) and Stadlbauer et al., (2020) with modifications. […] Despite the difference in sample type and handling, we are confident these do not impact on antibody levels or ELISA signal from our own in-house comparisons (data not shown) as well as data from others (Amanat et al., 2020).”

3) It's not clear how the presence of *P. aeruginosa* is associated with the disease severity or pneumonia. One of such confusing statement is "There were no further concerns about a bacterial infection during his ICU stay although a third *P. aeruginosa* was cultured from a sample (ETT- 3) collected two days after he completed his course of meropenem, that was resistant to both pip/taz and meropenem (MICs >16 mg/l and >8mg/l respectively)" in the Introduction. Does this mean in ETT-3, this bacterium is still detected and resistant to antibiotics? But why it becomes not a concern? If it's because the patient is in his remission stage, it makes people wonder if the earlier concern from the bacterial infection during ETT-1 and ETT-2 is a side effect from the severe SARS-CoV2 symptoms instead of any causation or association effects.

The diagnosis of VAP in intubated patients is complex. In our intensive care unit, there is a daily infection review led by the Consultant Medical Microbiologist along with a senior Pharmacist. At these meetings, the clinical progress of all patients, inflammatory indices (WCC, CRP, and procalcitonin) and the relevance of concurrently cultured isolates is discussed. For this patient, the initial samples ETT-1 and ETT-2 coincided with an increasing oxygen requirement, fever, and the patient becoming more clinically unwell, and therefore a clinical diagnosis of likely VAP was made and patient was treated with antibiotics for same. At the time of ETT-3, the patient was clinically well and did not meet any clinical diagnostic criteria for VAP, so this was not deemed significant. We explain this in greater detail in the revised manuscript (Introduction).

4) How's the level of *P. aeruginosa* after the patient is discharged from hospital and even before the development of SARS-CoV-2? Although it's too late to ask such a question about the level before SARS-CoV-2, is it possible that *P. aeruginosa* is an existing bacterium in this patient other than from a secondary bacterial infection after SARS-CoV-2?

Measuring levels of bacterial load in VAP is extremely difficult and not routinely done in clinical practice. *P. aeruginosa* is a ubiquitous environmental bacteria, and while it can be found as a commensal in the respiratory tract of some individuals, this is typically only when they have some underlying chronic suppurative lung disease such as bronchiectasis. Prior to his COVID-19 diagnosis this relatively young patient was healthy with no prior medical issues, which makes us think it is more likely the positive cultures and metagenomic data represent hospital or ICU acquired colonisation and subsequent infection of this patient. We have made a comment to explain this in the manuscript (Results section).

5) What is the logic of performing sequencing on ETT-1 and ETT-3 but not ETT-2? ETT-2 genome sequencing might help to show some evidence of evolution of the mutation points.

There is no logic to this, just an unfortunate situation that arose due to the pressures the diagnostic lab was under during the first wave of the pandemic. ETT-2 was not made available for research and likely disposed of in error, and this has been explaining in the revised manuscript (Results section).

6) During metagenomics, did the authors see other microbes? It's surprising if there are only two bacterial species detected in those samples using metagenomic sequencing. Or is it because these two strains are the most dominant ones that detected? If so, it would be useful to show the whole microbial community profile from this sequencing.

In the original manuscript we only reported the whole genomes we were able sequence from the clinical samples. However, as correctly pointed out by the reviewer, the omission of information on what other organisms were present is unhelpful. In the revised manuscript we now described these data (Figure 2B).

7) Since there are two strains that are fully sequenced, although Staphylococcus haemolyticus is "not recognized as lung pathogens" as stated by the authors, if this particular strain is resistant to methicillin, would it also be a potential target strain with AMR deteriorating the disease in this scenario?

Having revisited our metagenomic data we found a labelling error such that the *S. haemolyticus* genome we originally reported as being part of ETT-1 was in fact part of ETT-3. Given the relatively healthy status of the patient at the time ETT-3 was collected and the likelihood that it was a commensal as opposed to an infecting organism, we have removed specific mention of this from the revised manuscript. Instead, we report its presence as part of the microbial make-up of ETT-3 in Figure 2B.

Did the authors also detect it from ETT-3? Would it be necessary for the authors to carry out a similar assay as Figure 3E and f using the S. haemolyticus antigen to show if it has an effect?

As mentioned above, when we revisited our metagenomic data we found a labelling error such that the *S. haemolyticus* genome we originally reported as being part of ETT-1 was in fact part of ETT-3. In the revised manuscript we now include greater details on the microbial composition of both ETT-1 and ETT-3, where the only bacterial DNA found in ETT-1 was that of the infecting P. aerunginosa strains; whereas ETT-3, where the patient had recovered from his VAP there was greater microbial diversity, including the *S. haemolyticus* strain. As such we don’t believe it to be necessary to perform ICS experiment with a *S. haemolyticus* antigen.

8) It's very interesting observation that CD4^+^, CD8^+^ and TCR-gd T-cells appear to wane between days 23-28 and boosted after day 28, is there an antibiotic effect here as pip/taz was administrated ~day 28? How to explain a decrease around day 45 on Figure 3B? Was that also an antibiotic effect by the starting of meropenem? It could be further discussed in the paper. And authors should explain a little more in the text about the difference between Figure 3A and 3B.

The initial waning of T-cell activation and proliferation shown in Figure 3E-F could suggest a declining response to a viral infection, which would be in line with another case report that looked at T cell kinetics longitudinally in a COVID-19 patient and shows that T-cell activation during SARS-CoV-2 infection declines from around 20 days post-infection (Theravajan et al., 2020). While there is some evidence that antibiotics can interact with aspects of host immunity, the available data suggests the activity is typically suppressive. With this in mind, we feel the increase in activity that coincided with the administration of pip/taz is more likely due to the response of the immune system to the VAP. Regarding the decrease following meropenem treatment, it is possible it plays a roll in the observed decrease in proliferation of some of the cell types we studied (i.e. the CD8^+^ and gd T-cells) but as this was not observed for the other cells type studied, we feel it is too speculative to bring into this descriptive study.

Reviewer #3:Dr Gregorova and colleagues present an interesting case report entitled "Post-acute COVID-19 associated with evidence of bystander T-cell activation and a recurring antibiotic-resistant bacterial pneumonia." The report details a patient that presented with respiratory failure 20 days after testing positive for COVID-19 on screening following exposure to a family member. The patient's course was complicated by two episodes of VAP with metagenomics serially revealing complete genomes of *P. aeruginosa* encoding antibiotic resistance. The fundamental thrust of the paper is the inference that both the initial respiratory failure and the subsequent VAP episodes result directly or indirectly from immunopathology driven by bystander T cell activation, which the investigators assess by determining CD4 and CD8 cell cytokine expression following PBMCs exposure to viral and bacterial peptides.The report is easy to read and proposes an explanation for the patient's respiratory failure that is congruent with known immune responses to SARS-CoV-2. Indeed, as the authors note, this may help explain the observed efficacy of dexamethasone in patients with respiratory insufficiency. However, there are limitations that reduce enthusiasm.1) The paper broadly suggests/implies that bystander activation of T cells is a proximate cause of the VAP episodes without any direct evidence to support this notion. It is well established in the literature that post-viral leukocyte dysfunction (most often lymphocytes and monocyte/macrophage lineage cells) enhance susceptibility to secondary bacterial infections. In this case, the authors demonstrate variably elevated numbers of T cells but do not demonstrate failures of *P. aeruginosa* sensing/responding/clearance by any cell population.

As this is a case study with a single patient, we were very careful not to make any statements or infer any specific thoughts we had on what has happened with this patient. It’s always possible that such inferences slip in inadvertently, and if the reviewer can direct us to where these are to be found we will modify/temper that language used.

It is interesting that the reviewer felt we were inferring the bystander activation was a proximate cause of the VAP, as we believe the opposite is more likely to be true. With or without prior viral infection, the mechanical ventilation of patients leaves them highly susceptible, such that between 10 and 30% will typically succumb to a bacterial VAP, and this patient has fallen into this category. If we were pushed to make a statement on what we believe might be going on (given the limitation of a single patient) it’s that the onslaught of an invasive bacterial infection immediately after the viral infection (and the heightened and prolonged immune response this triggered) is responsible for the observed bystander activation. This activation may be responsible for the ongoing inflammatory associated problems that patient is experiencing, but again, we believe we have been very careful not to be explicit about this in the manuscript, give its limitation as a case study.

It is worth noting here that although we detect both an antibody and T-cell response specific to *P. aerunginosa*, we make no comments or assertion as to their role in the clearance of the VAPs, as in both cases the patient was treated with appropriate antibiotics, which will have had a dominant affect on clearing these infections.

2) The essential arguments for bystander activation are the stimulation experiments in Figure 3E-F. These data are widely variable and are difficult to interpret in the absence of data from the times that VAP was first suspected. The only shown dates are from ICU admission and two dates each after several days of VAP therapy.

Unfortunately, we were unable to perform T-cell stimulation and ICS on samples from all time points from this patient, due to the limited blood samples that we could ethically collect from a severly ill patient. As explained above, we have repeated the T-cell stimulation with CMV and SARS-CoV-2 spike peptides on the last vial of PBMCs made available to us, and we now also include data (in duplicate) on the T cell response to both CMV and SARS-CoV-2 (Figure 4—figure supplement 1). We were also able to measure the expression of activating and proliferating markers on CMV-specific T cells and show that CMV-specific CD8^+^T cells are activated and proliferating in line with other studies on bystander activation, showing increased expression of activation and proliferation markers by CMV-specific CD8^+^ cells when compared to the CMV-specific CD4^+^ cells. We also found that the CD8^+^ CMV-specific T cells were predominantly T_EMRA_ and T_EM_ cells while CD4^+^ CMV-specific T cells were predominantly T_EM_ cells (Figure 4F,I), which is in line with previous reports on this phenomenon.

3) In a single case report, it is always challenging to confidently know what is driving an observed immune response. This is particularly so in a patient who was well at the time of viral diagnosis but presents 20 days later. Certainly, the authors' inference that the respiratory failure represents COVID-19 immunopathology is plausible, even likely. However, since the interval between diagnosis and respiratory failure is longer than the mean interval reported in recent large series and since longer intervals provide more opportunities for patients to acquire new pathogens/be exposed to new insults, more robust evidence than a single case would typically be expected to make such a claim.

As discussed above, we were careful not to make any claims in relation to what is driving the observed immune response, given the limitation of the study. However, if the reviewer can point us to where these are inferred we will be more than happy to address/temper these.

We are however confident that the SARs-CoV-2 infection was responsible for this patient’s respiratory failure. There was a gap of 13 days from this patient testing positive to the development of symptoms, and a further week before his symptoms became such that he needed hospitalisation. The CDC, based on the findings of several studies, advise that the incubation period for COVID-19 can be as long as 14 days, which is a day longer than that experienced by this patient. (https://www.cdc.gov/coronavirus/2019-ncov/hcp/clinical-guidance-management-patients.html).

Also, although the patient tested negative for SARs-CoV-2 upon admission, none of his cultures up until ETT-1 grew anything to suggest a secondary infection. He also he had no other clinical features prior to the collection of ETT-1 to suggest any other clinical bases for his symptoms. The patient also had three sets CT scans take; upon admission to the ICU, when the first VAP was diagnosed and between his first and second VAP. The ground glass opacities typical of COVID-19 were evident throughout. These have now been included as supplementary data (Figure 1—figure supplement 1), as they are supportive of the fact that the patient required hospitalisation as a result of COVID-19.

4) The metagenomics data are impressively generated, but don't drive new understanding. That an intubated patient might acquire two related strains of *P. aeruginosa* during a single episode of critical illness is unsurprising. As above, there seems to be an underlying implication that the T cell activation drives the SNP differences between the isolates, but that is not further demonstrated. Metagenomic assessment at the time of ICU admission might have been supportive of the authors' assessment that there was no active viral infection (SARS-CoV-2 or otherwise) driving the acute decompensation that led to intubation.

As a study of a single patient, it’s difficult to say that what is presented drives any new understanding, but instead demonstrates the potential for what could be achieved/learned if applied to a wider cohort of patients. We include the metagenomics here as an illustration of what can be achieved above and beyond routine culture-based approaches, including the detection of AMR conferring mutations.

We are again interested to see that the reviewer has concluded that we felt the T-cell activity was driving the SNPs in the strains. This isn’t the case at all, and if the section in the manuscript that led the reviewer to think this could be highlighted to us, we will address the language we used there. Instead, we believe it is far more likely that the antibiotics used to treat the VAPs led to the selection of the AMR strains. Whether it is one strain that acquires the mutations to become resistant, or sequential infections with a different strain of *P. aerunginosa* with differing AMR profiles is impossible to say, but we discuss both of these scenarios in greater detail in the revised manuscript (Results section).

[Editors’ note: what follows is the authors’ response to the second round of review.]

The case study's value is in suggesting that bystander T cell activation in SARS-CoV2 infection may have led to a severe COVID-19 presentation. This supports the benefit of corticosteroids in COVID-19. The message is indeed plausible and important. The authors need to tell us how much it adds to what we know (see Mathew et al., 2020).

The Mathew et al., 2020 paper describes distinct immunophenotypes that associate with disease in 125 hospitalized COVID-19 patients at a single timepoint. There is information within that paper describing some COVID-19 patients that acquired secondary infections (e.g. Figure S2E) but little information regarding the nature of the secondary infection, and only partial immunophenotyping for these patients is reported.

Our current case report, which was submitted prior to publication of the Mathew et al., study, provides a detailed and in-depth longitudinal analysis of the kinetics of immune activation of conventional T cells, gd T cells and other immune cells during a secondary bacterial infection post COVID-19, albeit in a single patient. We also show evidence of bystander activation of T-cells, which was not examined in the Mathew et al., paper. In the revised manuscript we now discuss the parallels between our work and what was reported in the Mathew et al., paper (Results section and Discussion section).

The essence of the case report seems to be that a patient with asymptomatic SARS CoV2 infection developed respiratory symptoms suggestive of COVID19 at 2 weeks post-infection, requiring hospitalisation. After more than 96 hrs of hospitalisation the patient required ICU admission and was SARS CoV2 negative, but resistant P. Aeurginosa positive on ETT aspirate. At this point bystander T cell activation was noted. The authors link the bystander activation to the SARS-CoV2 infection and disease progression, which is plausible but uncertain due to questions: namely (a) whether an additional infection may have occurred in 2 weeks, (b) whether bystander activation measured at 3 weeks is confidently determined and relatable to original infection, and (c) whether the hospital-acquired multi drug resistant pneumonia is related to the immune dysfunction.a) Is this severe COVID-19 or a new insult given that 15 days passed from PCR positivity for onset of symptoms? A clear comparison of this to reported intervals between onset of infection and clinical deterioration would help make the case stronger. If this is very unusual, then alternative explanations, such as a new infection, become more likely than severe COVID-19. Clinical data would be helpful, including any repeat PCR at admission.

There were 13 days between the patient testing positive for SARS-CoV-2 and developing symptoms. Once admitted to our intensive care unit he was assessed, as all our patients are daily, for signs of additional infection through a process led by a consultant medical microbiologist along with senior pharmacist. At these meetings, the clinical status of the patient, inflammatory indices (WCC, CRP, and procalcitonin) and any culture results are discussed. We had no concerns at the time of admission of this patient to the ICU that he had any infection other than that of SARS-CoV-2. We agree that the time between him testing positive by RT-PCR and develop symptoms is atypical, however the CDC, based on the findings of several studies, advise that the incubation period for COVID-19 can be as long as 14 days, which is a day longer than that experienced by this patient. (https://www.cdc.gov/coronavirus/2019-ncov/hcp/clinical-guidance-management-patients.html). We have discussed this in greater details within the revised manuscript (Introduction)

With regards to additional clinical data, he was tested for SARS-CoV-2 upon admission to hospital as well as upon admission to the ICU the following day, both tests were negative. Until ETT-1, none of his cultures grew anything to suggest a secondary infection. The patient had three sets of CT scans take; upon admission to the ICU, when the first VAP was diagnosed and between his first and second VAP. The ground glass opacities typical of COVID-19 were evident throughout. These have now been included as supplementary data (Figure 1—figure supplement 1).

b) *P. aeruginosa* infection is probably hospital-acquired and multiple strains are quite possible. The link to bystander cell activation only arises if it led to ineffectual sensing/clearance. Can this be supported by any experimental evidence beyond clinical deterioration?

If we have understood this question correctly, we are being asked if we believe there to be a link between the clearance of the recurring VAPs and the observed bystander activation; and whether we have any additional experimental evidence to support this? While we provide additional data (described below) in support of the observed bystander activation, we have been careful not to make any causal links between what we have observed for this patient, except to propose that this massive level of immune activation may be contributing to his ongoing inflammatory-associated symptoms. With regards to clearance of the VAPs, having been treated with antibiotics, the effect of the immune system is likely to be minimal, so we believe it unlikely that the bystander activation contributed positively or negatively to this. We have ensured within the revised manuscript that this is made clearer

c) The essential arguments for bystander activation are illustrated in Figure 3E-F – that were found not to be particularly convincing. At the very least, please provide full data and experimental replicates.

As there are strict ethical constraints on how many blood samples can be collected from severely ill patients, we endeavoured throughout this study to make the best use of what we were able to obtain, which was what was presented in the original manuscript. However, we have since managed to obtain one final vial of PBMCs from this patient collected during his time in the ICU at day 38 post RT-PCR+ test. To address the concern raised here, we have used these cells to provide replicate data that shows the presence of CMV-specific CD4^+^ and CD8^+^T cells within the sample. We also show new data that demonstrates (in duplicate) that these CMV-specific CD4^+^ and CD8^+^T cells were activated and proliferating, based on their expression of activation and proliferation markers, which supports that they are bystander activated (Sandalova et al., 2014; Rivino et al., 2015) (Figure 4D-E,G-H and Figure 4—figure supplement 4A-B). The expression of activation and proliferation markers by CMV-specific CD8^+^T cells during an infection with an unrelated pathogen is in line with other studies on bystander activation, showing increased expression of activation and proliferation markers by CMV-specific CD8^+^ cells when compared to the CMV-specific CD4^+^ cells. We also found that the CD8^+^ CMV-specific T cells were predominantly T_EMRA_ and T_EM_ cells while CD4^+^ CMV-specific T cells were predominantly T_EM_ cells (Figure 4F,I), which is in line with previous reports on this phenomenon.